# Effects of In Ovo Supplementation with Nanonutrition (L-Arginine Conjugated with Ag NPs) on Muscle Growth, Immune Response and Heat Shock Proteins at Different Chicken Embryonic Development Stages

**DOI:** 10.3390/ani10040564

**Published:** 2020-03-27

**Authors:** Sivakumar Allur Subramaniyan, Darae Kang, Sharif Hasan Siddiqui, Jinryong Park, Weishun Tian, Byungyong Park, Kwanseob Shim

**Affiliations:** 1Department of Animal Biotechnology, College of Agriculture and Life Sciences, Jeonbuk National University, Jeonju 54896, Korea; sivaphdbio@gmail.com (S.A.S.); kangdr92@gmail.com (D.K.); HASANSHUHIN@gmail.com (S.H.S.); wlsfyd1321@naver.com (J.P.); 2College of Veterinary Medicine and Bio-Safety Research Institute, Jeonbuk National University, Iksan 54596, Korea; tianws0502@gmail.com (W.T.); parkb@jbnu.ac.kr (B.P.)

**Keywords:** nanoparticles, silver, L-arginine, conjugation, muscle growth

## Abstract

**Simple Summary:**

In this study, we have analyzed the effects of *in ovo* supplementation of inorganic and organic synthesized silver nanoparticles (Ag NPs) conjugated with L-arginine (L-Arg) that were injected in chicken embryo at three different embryonic development stages (8th d, 14th d and 18th d). We investigated the effects of both conjugated Ag NPs on the survival, hatchability and body weight on hatching day (1 d old chicks). The expression of muscle growth related proteins, mainly myoblast determination protein (myoD), myogenin and heat-shock proteins (HSPs) were analyzed in pectoral muscles. The serum level of immunoglobulin M (IgM), serum glutamate oxaloacetate transaminase (SGOT) and serum glutamate pyruvate transaminase (SGPT) were also examined. Our study shows that 14th day of injection of both types of conjugated Ag NPs promoted survival rate and hatching rate at 8 d and 18 d from injection. The immunoglobulin (IgM) levels in serum and the expression of muscle growth related proteins (myoD and myogenin) were dramatically improved; in addition, HSP-60 and HSP-70 expression were declined at 14th d of injection. The serum levels of SGOT and SGPT were decreased at 14d injection when compared to 8 d and 18 d injection. Hence, the 14th day would be suitable day for injection of L-Arg with Ag NPs to promote the survival rate, hatching rate, immune system and muscle growth. Our results illustrated that plant mediated synthesis of Ag NPs at 1000 μg and chemically synthesized Ag NPs at 100 μg concentration would be a better choice to make nanonutrition with L-Arg (100 μg) to carry the nutrients without any toxicity at 14 d injection.

**Abstract:**

The aim of the study was to analyze the *in ovo* injection of inorganic and organic synthesized silver nanoparticles (Ag NPs) using *Brassica oleracea* L. var. capitate *F*. *rubra* (BOL) conjugation with L-Arginine (L-Arg) on the immune, muscle growth, survivability and hatchability of broiler chickens. The conjugation of L-Arg (100 μg) with 1000 µg of Ag NPs synthesized by (BOL)-extract and L-Arg (100 μg) conjugated with 100 µg of Ag NPs inorganic synthesized were injected into fertile eggs at 8 d, 14 d and 18 d of incubation. Survival and hatching rate were significantly improved in the dose of L-Arg (100 μg) with 1000 µg (BOL-Ag NPs) and L-Arg (100 μg) with 100 µg (C-Ag NPs) on 14 d injection whereas it was decreased on 8 d or 18 d injection. Moreover, the protein expression of muscle development markers such as myogenin and myoD were significantly uprelated in 14 d of incubation whereas the heat shock proteins (HSPs), such as HSP-60 and HSP-70, were significantly upregulated in 18 d incubation. In addition, the liver function marker of serum glutamate oxaloacetate transaminase (SGOT) and serum glutamate pyruvate transaminase (SGPT) were significantly decreased and the immunoglobulin (IgM) levels were increased in a 14 d incubation period in serum at the same concentration.

## 1. Introduction

Nutrition by *in ovo* injection is a new methodology for directly supplying additional nutrients and bio-active compounds, mainly amino acids, and carbohydrates into the fertilized embryo particularly during incubation time to promote the growth and development of muscles during embryogenesis. Moreover, the hatching rate and pectoral muscle growth were increased by nutrients by *in ovo* injection [1]. This early nutritional strategy offers the promise of sustaining progress in production efficiency and welfare of commercial poultry. Recent study reported that, metal nanoparticles (NPs) such as silver, zinc, copper and selenium do not affect the embryo development and hatchability and they improve the immune system [2]. Few studies have been conducted to comprehensively evaluate the roles of amino acids such as leucin, ascobilc acid, methionine-cysteine, sulfur amino acids, vitamin C and probiotics and whether they can improve the growth performance and immune status of chicken. Certainly, glutamine conjugated with Ag NPs improved the growth and immunity status of embryos and chicks [3].

The administration of amino acids into fertilized broiler eggs via *in ovo* feeding has provided poultry companies with an alternative method to increase hatchability and muscle growth weight of newly hatched chicks [4,5]. L-Arginine (L-Arg) is classified as an essential amino acid and it has been reported that *in ovo* administration of L-Arg in the embryonic phase could increase the growth rate and muscle mass for the reason that L-Arg can stimulate the release of growth hormone [6,7]. L-Arg (100 μg/μL/egg) injection at 14th day of chicken embryo increases chick body weight, hatch rate, muscle growth-related proteins and stimulates the immune response (IgM) [8].

The therapeutic uses of silver nanoparticles (Ag NPs) countermeasures against deadly disease and the prevention of infectious avian influenza viruses which can spread rapidly to poultry flocks. They need further research, particularly with regard to their effects in biological systems. Recent studies have shown that Ag NPs penetrated livers of chickens and reduced the activity of the liver enzymes ALP and LDH, based on their physical activity, surface ratio and chemical stability. Previous reports stated that *in ovo* supplementation of Ag NPs could promote growth performance and immune system of broilers and layers [9]. Plant mediated synthesized Ag NPs are superior choices to other amino acids or proteins with the aim to produce a new method of *in ovo* nanonutrition that avoids the toxicity of inorganic synthesized Ag NPs [9]. *In ovo* supplementation of Ag NPs conjugated with glutamine (100 mg/L) penetrate tissues and cells can improve the growth and immune status of embryos and chicks. Furthermore, Ag NPs conjugated with hydroxyproline can enhance the development of blood vessels [10]. *In ovo* injection of Ag NPs complex with amino acid can enhance both innate and adaptive immunity in chicken [11].

Muscle development is mainly determined during embryogenesis and the final number of muscle fibers is accomplished in prenatal and early post hatch periods [12]. Moreover, muscle maturation during embryogenesis is dependent on the development of a vessel network, which provides cells with oxygen and nutrients.

Administration of L-arginine at a dose of 0.7% in turkeys and 1.0% in quail by *in ovo* feeding could increase body weight and post-hatch performance [13]. Up to date, there have been no attempts to address obstacles that impede the manipulation of stages of chick embryos *in ovo* by treatment with conjugated silver nanoparticles (Ag NPs) with L-Arginine (L-Arg). Therefore, the aim of the present study was to investigate the effect of *in ovo* injection of conjugated L-Arg with inorganic and organic synthesized Ag NPs to embryo at different incubation time (8^th^ day (d), 14th day and 18th day) on hatchability, survivability, body weight, muscle growth-related proteins such as myogenin and MyoD and immunoglobulin M (IgM) levels in 1-d-old chicks.

## 2. Materials and Methods

### 2.1. Ethics Statement

The experimental protocol was approved by the Institutional Animal Care and Use Committee (IACUC) with the approval from ethical committee of Jeonbuk National University, Korea Republic (JBNU 2015048).

### 2.2. Chemicals

Silver nanoparticles (Ag NPs) and L-Arg, were obtained from Sigma-Aldrich (Sigma-Aldrich, St. Louis, MO, USA). Antibodies were purchased from ENZO Life Science (Farmingdale, NY, USA). Chemiluminescence for band detection was bought from Thermo Scientific (Rockford, IL, USA). All laboratory glassware was acquired from Falcon Labware (Becton, Dickinson and Company, Franklin Lakes, NJ, USA).

### 2.3. Synthesis of Ag NPs Using BOL Extracts

Ten grams of fresh and healthy *Brassica oleracea* L. var. capitate *F. rubra* (BOL) leaves were cleaned with tap water, followed by cleaning with distilled water (D.H_2_O) several times to remove any external particles adhered onto their surface. They were then boiled in 100 mL of D.H_2_O for 5 min in a microwave oven by the method of [14]. The resulting extract was filtered through a Whatman^®^ filter paper. The reaction mixture containing 6 mL of extract and 44 mL of D.H_2_O was added to 1 mM AgNO_3_ to a final reaction volume of 50 mL and kept in dark place for four hours at room temperature. A control setup with only leaves’ extract was also maintained throughout the experiment.

### 2.4. Synthesis of Ag NPs’ Composites Using Polyvinylpyrrolidone (PVP)

For the synthesis of Ag nanoparticles, a Sharp microwave oven (model R-259) was used. In a typical procedure, 10 mL of 1% (*w*/*v*) ethanolic solution of PVP and 0.2 mL of 0.1 M AgNO3 in a 25 mL closed conical flask were placed in a microwave oven operating at 100% power of 800 W with a frequency of 2450 MHz for 5 s. The colorless solution instantaneously turned into a characteristic pale yellow color, indicating the formation of Ag NPs. The advantage of a microwave-mediated synthesis over conventional heating is that it has improved reaction kinetics, generally by a magnitude of one or two due to rapid initial heating and the generation of localized high-temperature zones at reaction sites [15].

### 2.5. Characterization of Ag NPs

Preliminary characterization of Ag NPs done through visual observation for change in color. An aliquot of the reaction mixture was analyzed using UV-Vis spectroscopy (UV-1800, Shimazdu Corp., Kyoto, Japan) in wavelength range of 300 to 700 nm. The sample was centrifuged (15,000 rpm for 20 min) the supernatant was discarded and the pellet was collected. For electron microscopic studies 25 µl of the Ag NPs was drop coated on a copper grid and the images of NPs were studied using a high-resolution transmission electron microscope (HR-TEM) (S-4800, Hitachi, Japan). For X-ray diffraction (XRD) studies, dried NPs were coated on XRD grid and the spectra were recorded using powder X-ray diffractometer (D/Max 2500, Rigaku, Japan) and the resulting powder was used for further examinations.

### 2.6. Experimental Design and Incubation

In this study, fertile broiler eggs (1000) of *Ross*^®^ 308 were obtained from Samhwa-Won Jong, South Korea. The eggs were numbered and weighed (60 ± 1.36 g) individually; abnormally weighted eggs were discarded from the experiment. The eggs were candling on 8th d for checking living embryos, the nonfertile eggs were discarded from the incubator at the 8th d of incubation. The 880 fertilized eggs were selected and randomly divided into ten groups (4 × 20 × 3 = replication × eggs × injection) (Table 1). On day 8, 14 and 18 of incubation, the blunt (air space/air sac) side of the egg was determined with a marker pen and disinfected (sterilization) with cotton swabs with 70% ethanol. After that *in ovo* administration of L-Arg (100 µg/100 µL/egg); conjugation of L-Arg (100 µg) with BOL extract synthesized Ag NPs (1000 µg/100 µL/egg); conjugation of L-Arg (100 µg) with chemical synthesis Ag NPs (100 µg/100 µL/egg) on 8th, 14th, and 18th day, respectively, through the air sac of the eggs using a 21-gauge needle. Immediately after the injection, the hole was sealed with liquid paraffin. Eggs were then placed in an incubator for 18 days under standard conditions (temperature of 37.8 °C; humidity of 60%). Eggs in the setting compartment automatically turned 98 °F every 3 h (eight times a day). On the 18th day of incubation, eggs were moved to hatching boxes promptly placed in a hatching incubator with humidity maintained at 60% and temperature set at 37 °C. Egg weight, chick weight, chick weight to egg weight ratio and hatchability were recorded on 21 d.

### 2.7. Survival Rate Measurement

Embryos’ survival rates during the incubation period were measured on the eighth day. After *in ovo* injection, the number of live eggs were recorded from the total number of each treatment group on 18d. The percentage of survival rate was calculated with the following Equation (1):(1)$$\mathrm{Survival}\;\mathrm{rate}\;\%=\frac{\mathrm{No}.\;\mathrm{of}\;\mathrm{live}\;\mathrm{eggs}}{\mathrm{No}.\;\mathrm{of}\;\mathrm{fertilie}\;\mathrm{eggs}}\times 100$$

### 2.8. Hatching Rate and Body Weight Measurements

On the 21st day, hatched chicks were moved from the hatcher incubator to hatching boxes to determine hatching rates. The hatched chicks were kept without feed and water at 32 °C and then the groups were weighed to record their live body and liver weight. The hatching rate was calculated with the following Equation (2):(2)$$\mathrm{Hatching}\;\mathrm{rate}\;\%=\frac{\mathrm{No}.\;\mathrm{of}\;\mathrm{chicks}\;\mathrm{hatched}\;\mathrm{on}\;21\mathrm{st}\;\mathrm{day}}{\mathrm{No}.\;\mathrm{of}\;\mathrm{fertilie}\;\mathrm{eggs}\;\mathrm{that}\;\mathrm{were}\;in\;ovo\;\mathrm{fed}}\times 100$$

### 2.9. Biochemical Indices

At the end of the experimental period, three hatched chicks of each replicate group per treatment were randomly selected and then sacrificed under anesthesia (diethyl ether) for sampling. Blood was collected from the jugular vein into tubes for serum separation. The breast muscle was collected and washed in ice-cold saline. It was then homogenized with 0.1 M of cold phosphate buffer, pH 7.4. Assays were done for serum and liver. Concentrations of serum glutamate pyruvate transaminase (SGPT) and serum glutamate oxaloacetate transaminase (SGOT) were measured using commercial kits (Asan Pharamaceuticals Co., Ltd., Seoul, Korea) following the manufacture’s specification.

### 2.10. Measurement of IgM Concentration in Serum

Serum samples (4 μL) were collected (dilute samples 1:60,000 into 1× Diluent N) from three hatched chicks of each replicate group per treatment to determine serum immunoglobulin (Ig) M levels using chicken IgM ELISA kit (Abcam, Suite B2304, Cambridge, MA, USA) following the manufacture’s specification. IgM levels were analyzed based on absorbance values measured at 450 nm.

### 2.11. Analysis of Heat-Shock Proteins (HSPs) and Muscle Related Markers by Western Blot

Proteins were extracted from 100 mg of muscle samples using radioimmunoprecipitation assay (RIPA) buffer to determine protein expression levels of HSP-60, HSP-70, myoD and myogenin in experimental groups. Protein concentrations were determined using a BIO-RAD protein assay kit (BIO-RAD). Extract samples containing 50 µg of protein were solubilized in *Laemmli buffer*, separated by 12% acrylamide gel and then transferred to Hybond-P PVDF membranes (GE Healthcare Inc., Amersham, UK) for 60 min at 200 mA. These PVDF membranes were blocked with 5% skimmed milk powder in 0.5 M of Tris-buffered saline (pH 7.4) with 0.05% Tween 20 (TBST) at room temperature for 2 h. The membrane was incubated with HSP-60 (1:2500), HSP-70 (1:2500), Myo-D (1:1000), myogenin (1:1000) and β-actin (1:2500) overnight at 4 °C overnight. After washing three times with TBST, these membranes were incubated with HRP-conjugated secondary antibodies (1:5000 dilutions) at room temperature for 60 min and then washed three times with TBST (10 min each wash). Protein bands were visualized using a Chemiluminescent assay kit from Thermo Scientific for 1–5 min. Bands were imaged with an iBright™ CL1000 Imaging System (Invitrogen in Thermo Fisher Scientific, Life Technologies Korea LLC, Jeonju-si, Jeollabuk-do, Korea) and quantified using Image J Software. The relative density of the band was normalized to that of β-actin as an internal control.

## 3. Statistical Analysis

Data for biochemical analyses are expressed as mean ± SD (*n* = 12). Statistical evaluation was carried out by Tukey’s HSD test post hoc following ANOVAs followed by Duncan’s multiple range test (DMRT) with the SAS^®^ software, version 9.4 (Institute of INC, North Carolina, USA).

## 4. Results and Discussion

### 4.1. Characteristics of Ag NPs

The aqueous BOL extract is pale pink in color. Addition of AgNO_3_ changed the color of the solution to reddish brown, within 2 h. UV–visible spectra of the solution exhibit absorption maxima at 430 nm for Ag NPs, which indicates the formation of Ag NPs by chemical and BOL extract. The HR-TEM images (Figure 1) of the Ag NPs reveal that they are well dispersed without much agglomeration. The Ag NPs are spherical in shape with an average size ranging from 5 to 40 nm (Figure 1). ICP-MS analysis reveals that the concentration of BOL-Ag NPs and C-Ag NPs is 514 and 628 μg/mg of powder, respectively.

### 4.2. FT-IR

Prior to Fourier-transform infrared (FT-IR) analysis, the samples were mixed with potassium bromide with appropriate ratio and compressed in a semi-transparent disk. Such samples were exploited for analysis. The FT-IR analysis was performed in the range between 4000 and 400 cm^−1^. L-Arg exhibits the peaks at 3426 cm^−1^ related to stretching vibrations of O-H. The combination of peaks positioned at 2913 and 2845 cm^−1^ are attributed to NH_3_^+^/C-H (asymmetrical bending) and NH_3_^+^ (torsional oscillation). The absorption band at 1650 cm^−1^ is ascribed to symmetric C=O bond. Amide stretching of L-Arg was found around 1318 cm^–1^. The stretching vibrations of C-C in backbone structure of L-Arg can be seen at 1225 and 1057 cm^–1^. Besides, the NH peak of L-Arg appears at 770 cm^–1^. FTIR spectrum of Ag nanoparticles obtained from BOL-extract synthesis shows that the peaks at 3550, 2926, 1635 and 797 are related to O-H, C-H, C-N and N-H, respectively. All of these peaks indicate that there is still some green extract present in Ag nanoparticles. L-arginine + Ag (BOL) shows the peaks of both L arginine and Ag NPs, indicating the successful combination of Ag and L-Arg (Figure 2).

### 4.3. XRD

XRD patterns of both chemical synthesized Ag and BOL synthesized Ag reveals the peaks at 2θ = 39.1, 43.2, 65.1 and 78.1 are found at (111), (200), (220) and (311) reflections of metallic Ag nanoparticles. In the case of green synthesized Ag, the additional peaks of green reducing agent were also observed. For the spectrum of L-arginine + Ag (BOL), both L arginine and Ag peaks were found, demonstrating the combination of Ag with L-arginine (Figure 3).

### 4.4. In Ovo Study

In the present work, we studied the effect of the conjugation of BOL-Ag NPs with L-Arginine or the conjugation of C-Ag NPs with L-Arginine by *in ovo* injecting them to three different embryonic stages on survival and hatching rates. Results are shown in Figure 4 and Figure 5. The survival rate was significantly increased in 2Tb and 2Tc (mentioned in Table 1) groups compared to other treatment groups. Results showed that the survival rate was different depending on the injection time of embryogenesis (8th, 14th or 18th day injection) with the same concentration (1000 µg for BOL-Ag NPs and 100 µg for C-Ag NPs). Ag NPs, glutamine (2.5 mg), and the complex of Ag NPs/Glu were nontoxic and did not affect the growth or the development of chicken embryo [12]. Furthermore, Ag NPs showed no harmful effects on the growth or development of embryos when these nanoparticles were used at concentrations below 100 μg/mL [12]. In addition, existing evidence demonstrated that on 14 d injection of L-carnitine, the hatchability was increased at 16d and 18d after injection. Nothing significant was observed in the *in ovo* L-carnitine injection (up to 8 mg dissolved in 100 μL of a commercial diluent) at the 18th day of incubation on hatchability of fertilized eggs in a young broiler breeder [16]. Moreover, the injection of chicken eggs with 100 μg/egg pyridoxine at 14 days of incubation period resulted in apparently higher hatchability than in uninjected control [17]. The 14th day incubation can be an appropriate time for improving the embryonic viability through the development of vessel network (embryo until day 15) [18]. The same mechanism might have occurred in our current experiment. In our experiment, the survival rate in the group injected on the 14th day of embryogenesis was not decreased after the injection. Instead, it was increased compared to that in the group injected on the 8th day or the 18th day of embryogenesis with the same dose of BOL-Ag NPs or C-Ag NPs.

Subsequently, we measured the hatchability rate. Results indicated that hatching rates were significantly different among groups. Significantly (*p* < 0.05) higher hatchability was recorded when both conjugated amino acids were injected on the 14th day, as compared to other groups. *In ovo* administration of NPa acting as bioactive agents and carriers of nutrients may be seen as a new method of nanonutrition [10]. Recent studies have shown that *in ovo* supplementation of Ag NPs [19], either alone or in combination with glutamine (25 mg/mL), can improve the growth and immunity status of embryos and chicks. The higher hatchability in the high immunity group after *in ovo* amino acid injection might be due to the availability of free amino acid through *in ovo* injection [20]. Such free amino acid might have stimulated embryonic gluconeogenesis which in turn helps hatching activities. Previous results showed that the hatchability of fertile eggs is not affected by *in ovo* L-carnitine administration (500 or 1000 μmol) at 18 days of incubation [21]. On the other hand, the dietary supplementation with L-carnitine (50, 100 or 150 mg) caused significant increases in egg fertility and hatchability with significant decrease of embryonic mortality on the 5th day of incubation [22]. *In ovo* injection on the 14th day of incubation could utilize amino acids [4]. These injected amino acids might have stimulated higher protein synthesis with lower protein degradation. Furthermore, Ag NPs conjugated with hydroxyproline can enhance the development of blood vessels [10]. Injecting the above two things on the 14th day of incubated embryo might have led to the development of blood vessels. The conjugated BOL-Ag NPs and C-Ag NPs might have travelled through these developed blood vessels to promote protein synthesis and improve hatching rate and survival rate. *In ovo* administration of Ag NPs nanoparticles upregulates the expression of fibroblast growth factor (FGF2) and vascular endothelial growth factor (VEGF) that are needed for satellite cell proliferation, differentiation, vasculogenesis and angiogenesis in tissues [23].

### 4.5. Body Weight

The concentrations of BOL-Ag NPs and C-Ag NPs were injected at three different time periods of embryogenesis. Body weights of hatched chicks in 2Tb and 2Tc groups were significantly (*p* < 0.05) higher than those of other groups (Figure 6). Body weights were low when conjugated amino acids were injected on the 18th day of the embryonic stage. Ag NPs with size less than 10 nm can penetrate into tissues and cells and localize inside cells [24,25]. *In ovo* administration of nanoparticles acting as bioactive agents and carriers of nutrients may be seen as a new method of nanonutrition [10]. Recent studies have shown that *in ovo* supplementation of Ag NPs [19], either alone or in combination with glutamine (25 mg/mL), can improve the body weight and immunity status of embryos and chicks. Hence, *in ovo* injection of Ag NPs with L-Arginine might have increased the body weight through Ag NPs that might have carried L-Arginine into the tissue and inside the cell to increase the body weight in 2Tb and 2Tc groups. Some investigations have demonstrated that Ag NPs do not have toxicity or affect the immune responses [26]. It has also been reported that Ag NPs, glutamine (25 mg/mL), and a complex of Ag NPs and glutamine do not affect embryos. However, the muscle percentage in the group treated with Ag NPs+glutamine is significantly increased compared to the group treated with Ag NPs alone [1]. Similarly, *in ovo* injection of Gly and Pro might have resulted in higher body weights of chicks in the 14th day injection group.

Ag NPs and the complex with glutamine improve nucleic acid synthesis and metabolic programing within cell, increasing their fibre area and consequently muscle mass [10]. The same mechanism could have occurred in our current study: the conjugated L-Arg could increase the number of nuclei per cell and by increasing fiber area, the growth performance would increase as well. Injection on the 14th day of the embryonic stage means that the amino acids can be utilized and protein synthesis is promoted [14]. These results suggest that the 14th day of injection could be a good time to promote the growth factor and increase the body weight of chicks.

### 4.6. Biochemical Indices (SGOT and SGPT)

Subsequently, we measured activities of hepatic enzymes (SGOT and SGPT) in blood serum as markers of functional and morphological states of the liver. Our results indicated that levels of SGOT and SGPT were significantly influenced by embryonic stages of injection for both BOL-AG NPs and C-Ag NPs. Biochemical indices (SGOT and SGPT) were significantly (*p* < 0.05) decreased in 2Tb and 2Tc groups compared to other groups which did not show significant effects of treatments (Figure 7). Previously, we have shown that treatment with BOL-Ag NPs (1000 μg) or C-Ag NPs (100 μg) did not have any toxic effect on the liver. Consequently, only SGPT levels were significantly increased when embryos were treated with C-Ag NPs (5000 μg). Thus, increasing the concentration of Ag NPs could increase levels of SGOT and SGPT in the blood which can lead to liver function damage. In fact, free radicals from Ag NPs can attack hepatocytes and release stored SGOT and SGPT to re-enter the blood serum [27].

Injection of Ag NPs into chicken embryo did not result in any negative changes in SGOT or SGPT levels, in agreement with previous results from experiments carried out *in ovo* [8]. Some studies have reported that the size, superficial coating effect, concentration of particles, surface charge, Zeta potential, composition and crystal form of the Ag NPs influence toxicity in embryos [28]. However, the toxicity of silver nanoparticles remains controversial. It is far from completely understood [29]. These inconsistent results appearing in the literature concerning the responsiveness to *in ovo* injection of L-carnitine might have resulted from many factors such as differences in strains and age of breeder hens, injection technique, site of *in ovo* injection, timing of injection (incubational age), dose and so on [30].

### 4.7. Measurement of IgM Concentration in Serum

At the 8th, the 14th or the 18th day of embryo stage (Figure 8), *in ovo* injection of BOL-Ag NPs (1000 μg) or C-Ag NPs (100 μg) was performed. At the 14th day injection, both BOL-Ag NPs and C-Ag NPs significantly increased the immune response measured by levels of IgM compared to the control and other stages (8th or 18th day). There was no significant difference in IgM level between the control and the 18th day injection groups (BOL-Ag NPs or C-Ag NPs treated groups).

Up to date, only a few studies have been conducted on poultry to evaluate the effect of nanosilver on immune and redox responses and lipid status of chicken blood [11,19,31]. *In ovo* feeding of amino acids can enhance growth-related genes and modulate the expression of immune genes in broilers. Moreover, *in ovo* administration of Ag NPs (15 mg) in combination with amino acids (threonine and cysteine at 15 mg) could improve the immune status of embryos. Thus, Ag NPs in combination with amino acid can act as a potential agent to enhance both innate immunity and adaptive immunity in chickens [32]. In our results, injection of BOL-Ag NPs (1000 µg) conjugated with L-Arg or C-Ag NPs (100 µg) conjugated with L-Arg (100 µg) on the 14th day improved the immunity by increasing IgM levels compared to other treatment groups.

### 4.8. Protein Analysis by Western Blot

Western blot was performed for muscle tissue to determine effects of L-Arg conjugated with BOL-Ag NPs (1000 μg) and C-Ag NPs (100 μg) injected at three different days of incubation period (8th day, 14th day and 18th day). Treatment altered protein levels of HSPs (such as HSP-60 and HSP-70) as well as Myogenin and Myo-D. As shown in Figure 9, protein expression levels of HSP-60 and HSP-70 were significantly (*p* < 0.01) downregulated in the following order: 3Tb < 2TC < 2Tb < 2C1 < 1Tc < 1Tb < 1C1 (Figure 9(ii)A,B) whereas expression levels of myogenin and Myo-D were significantly upregulated in the order of 2Tb > 2Tc > 1Tb > 1Tc > 3Tb > 3Tc (Figure 9C,D). Injection on the 17th day upregulated HSP-60 and HSP-70 when compared to 15-day incubation [33].

HSP-70 is a reliable index of stress in chickens while “3-hydroxyl-3-methyl-glutaryl coenzyme A reductase” has been used as an indicator of stress [34]. Pretreatment with L-Arg markedly reduced the dramatic downregulation of HSP-60 and HSP-70 in a hypoxic rat model.

The increased expression of HSP-60 and HSP-70 might be related to their leakages from tissue which may cause tissue injury due to free radical production [35]. Tissue injury can be caused by nitric oxide, a free radical, through stimulation of endothelial cells and neutrophils caused by a high dose of L-Arg. High dose of L-Arg after injection on the 18th day could increase the expression of HSP-60 and HSP-70 which can be involved in tissue injury due to free radical production. 3Tb and 3Tc groups might have shown the induction of tissue injury via the upregulation of HSP-60 and HSP-70, whereas the expression of HSP-60 and HSP-70 was downregulated in groups 2Tc and 2Tb in the order of 2Tc < 2Tb. Moreover, protein expression levels of myogenin and MyoD were significantly upregulated in 2C1, 2Tb, and 2Tc groups, whereas they were downregulated in 3Tb and 3Tc groups compared to other experimental groups. Oxidative stress can cause muscle degeneration (muscle atrophy) by reducing the myogenin and MyoD protein in skeletal muscles [36]. Some growth factors, such as oncogenes and cytokines can decrease the muscle mass by diminishing the activity of myogenin and MyoD, which is defined as muscle atrophy [37]. A previous study reported that myogenic regulatory factors, mainly MyoD and MRF4, are only expressed later in different embryonic muscle groups as a result of increased muscle mass [38]. Nevertheless, it has been proved that L-Arg conjugated with organic and inorganic synthesized Ag NPs could increase the muscle cell through the activation of myogenin and MyoD. Hence, the 14th day of injection of L-Arg (100 µg) + BOL-Ag NPs (1000 µg) could promote muscle growth better than the 8th or 18th day of injection.

## 5. Conclusions

*In ovo* injection of L-Arg with BOL and C-Ag NPs on the 14 day could promote the hatching rate, survival rate and immune response. Based on obtained results, it can be concluded that *in ovo* injection of L-Arg (100 μg) with 1000 µg (BOL-Ag NPs) or L-Arg (100 μg) with 100 µg (C-Ag NPs) is beneficial for hatchability when applied on day 14 of incubation. In addition, conjugation of L-Arg with 1000 μg of BOL-Ag NPs and 100 μg of C-Ag NPs injection within fertile eggs on day 14 of incubation is advantageous for subsequent performance. Additional studies on the effect of L-Arg (100 μg) with 1000 µg (BOL-Ag NPs) or L-Arg (100 μg) with 100 µg (C-Ag NPs) on the growth performance are currently under way in our laboratory.

## Figures and Tables

**Figure 1 animals-10-00564-f001:**
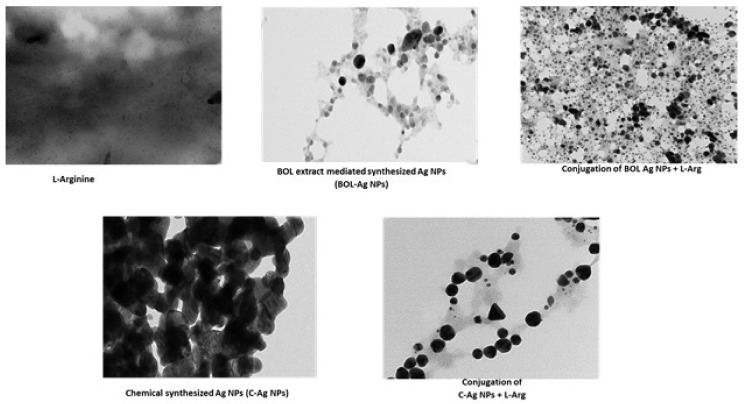
TEM images of Ag NPs (BOL-Ag NPs and C-Ag NPs), L-Arg and bio-complex of L-Arg with BOL-Ag NPs and L-Arg with C-Ag NPs.

**Figure 2 animals-10-00564-f002:**
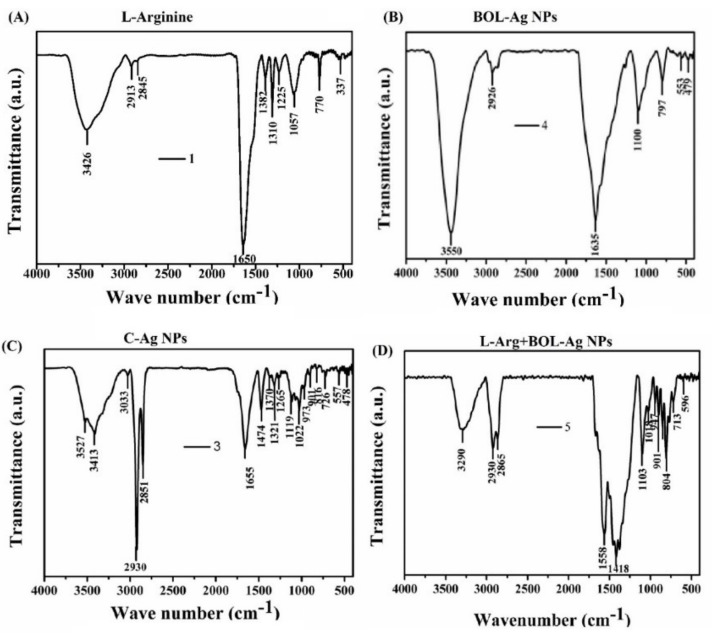
(**A–D**) Nature of functional groups and structure of the BOL extract and Ag NPs and conjugation of L-Arg with BOL-Ag NPs assessed by FT-IR spectra.

**Figure 3 animals-10-00564-f003:**
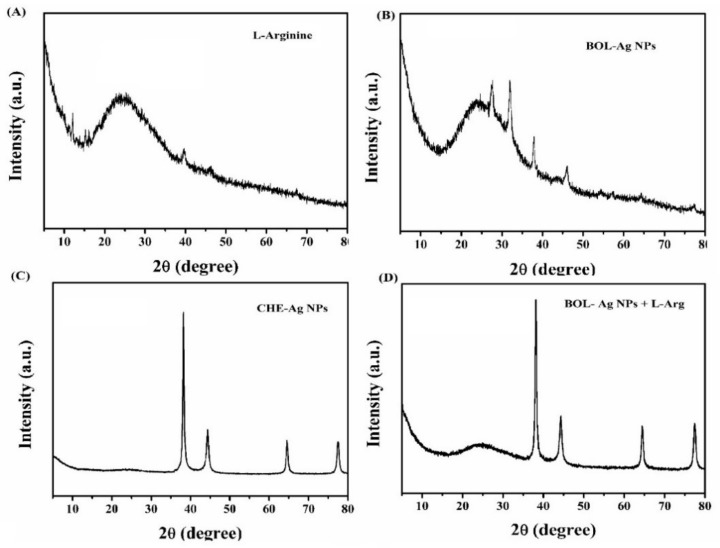
(**A**) Energy-dispersive X-ray spectroscopy profile of L-arginine; (**B**) biosynthesized (BOL) Ag NPs; (**C**) CHE-Ag NPs; (**D**) Conjugation of L-Arg with BOL-Ag NPs.

**Figure 4 animals-10-00564-f004:**
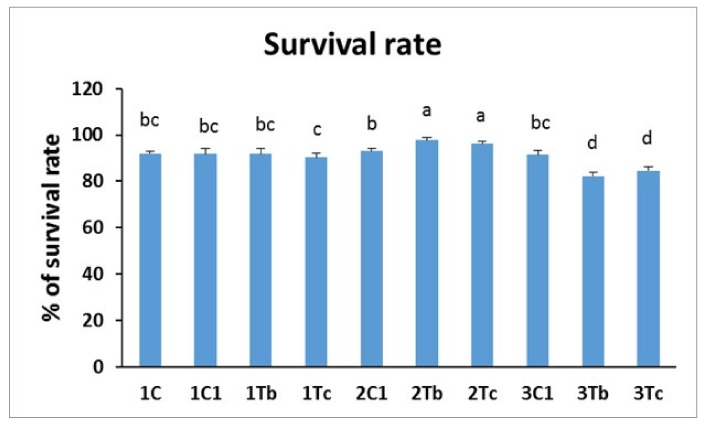
Effects of *in ovo* injections of L-Arg (100 μg) with 1000 µg (BOL-Ag NPs) or L-Arg (100 μg) with 100 µg (C-Ag NPs) at different developmental embryonic stages on survival rate. Small characters indicate significant differences among experimental groups at *p* < 0.05. Values are presented as mean ± SD from (*n* = 12) determinations.

**Figure 5 animals-10-00564-f005:**
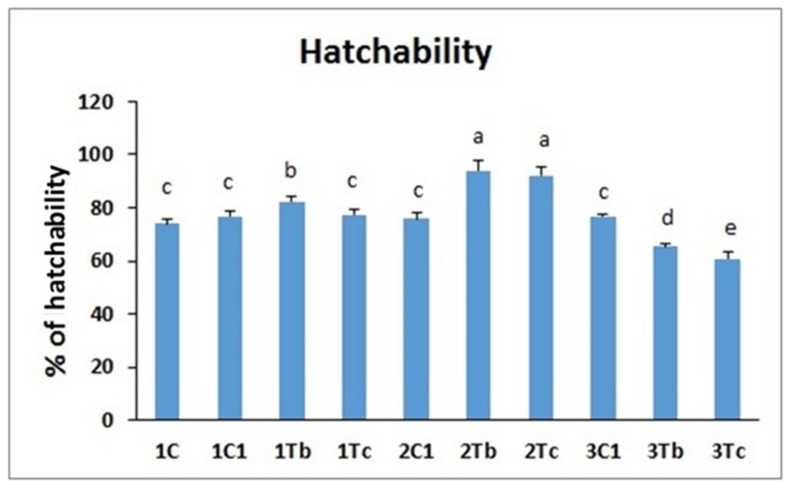
Effects of *in ovo* injections of L-Arg (100 μg) with 1000 µg (BOL-Ag NPs) or L-Arg (100 μg) with 100 µg (C-Ag NPs) at different developmental embryonic stages on hatching rate. Small characters indicate significant differences among experimental groups at *p* < 0.05. Values are presented as mean ± SD from (*n* = 12) determinations.

**Figure 6 animals-10-00564-f006:**
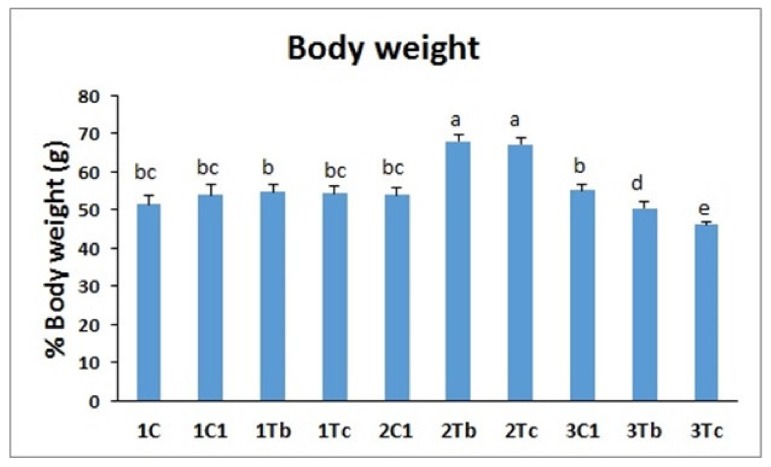
Effects of *in ovo* injections L-Arg (100 μg) with 1000 µg (BOL-Ag NPs) or L-Arg (100 μg) with 100 µg (C-Ag NPs) at different developmental embryonic stages on chick weight. Small characters indicate significant differences among experimental groups at *p* < 0.05. Values are presented as mean ± SD from (*n* = 12) determinations.

**Figure 7 animals-10-00564-f007:**
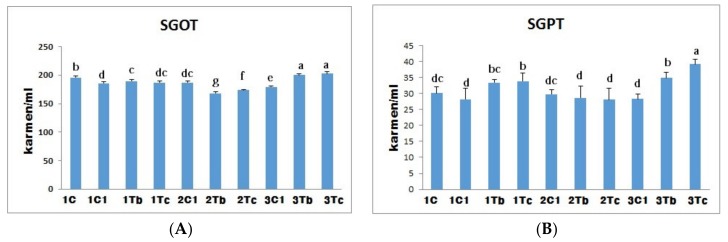
Effects of *in ovo* injections of L-Arg (100 μg) with 1000 µg (BOL-Ag NPs) or L-Arg (100 μg) with 100 µg (C-Ag NPs) at different developmental embryonic stages on serum glutamate oxaloacetate transaminase (SGOT) **(A)** and serum glutamate pyruvate transaminase (SGPT) **(B)** concentrations in serum. Small characters indicate significant differences among experimental groups at *p* < 0.05.

**Figure 8 animals-10-00564-f008:**
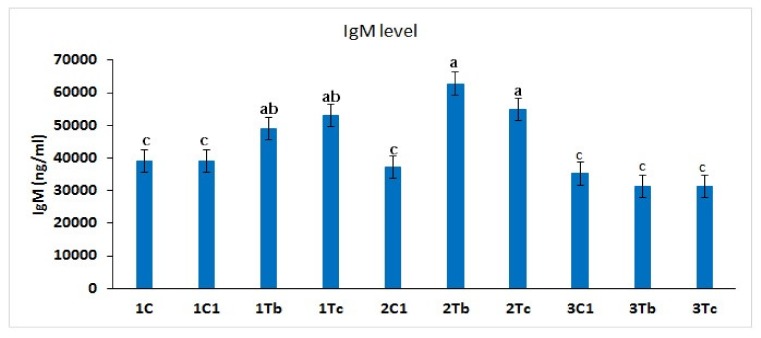
L-Arg induces IgM levels in different stages of chicken embryos. Small characters indicate significant differences among experimental groups at *p* < 0.05. Values are presented as mean ± SD from (*n* = 12) determinations.

**Figure 9 animals-10-00564-f009:**
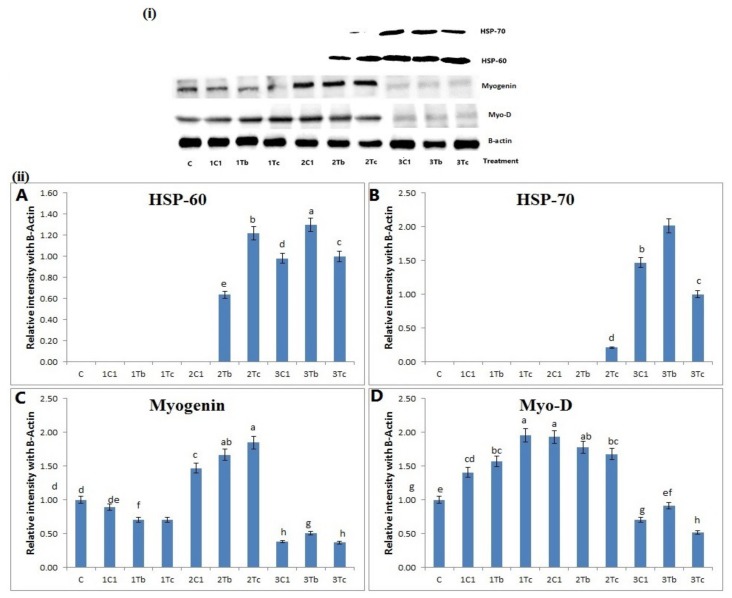
(**i**) Expression levels of L-Arg, HSP-60 and HSP-70 as well Myogenin and MyoD protein expression levels in different stages of chicken embryos after injection at different doses. Small characters indicate significant differences among experimental groups at *p* < 0.05. Figure 9 (**ii**) (**A**–**D**) bar graph represents quantitative expression of different proteins in all groups. Data are expressed as the ratio of relative intensity to the level of β-actin.

**Table 1 animals-10-00564-t001:** Experimental design for dose (L-Arg and conjugate with *Brassica oleracea* L. var. capitate *F. rubra* (BOL)-Ag NPs and C-Ag NPs) with different embryonic stage (8th day, 14th day and 18th day).

Group	Dosage	No. of Replication	Total No. of Eggs
1C	Control	4	80
1C1 (8th day)	PBS/100 µL/egg	4	80
1Tb (8th day)	100 µg (L-Arg) + 1000 µg (BOL-Ag NPs)/100 µL/egg	4	80
1Tc (8th day)	100 µg (L-Arg) + 100 µg (C-Ag NPs)/100 µL/egg	4	80
2C1 (14th day)	PBS/100 µL/egg	4	80
2Tb (14th day)	100 µg (L-Arg) + 1000 µg (BOL-Ag NPs)/100 µL/egg	4	80
2Tc (14th day)	100 µg (L-Arg) + 100 µg (C-Ag NPs)/100 µL/egg	4	80
3C1 (18th day)	PBS/100 µL/egg	4	80
3Tb (18th day)	100 µg (L-Arg) + 1000 µg (BOL-Ag NPs)/100 µL/egg	4	80
3Tc (18th day)	100 µg (L-Arg) + 100 µg (C-Ag NPs)/100 µL/egg	4	80

*In ovo* Injection and Treatment Groups.

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
