# Peer review of "Effects of In Ovo Supplementation with Nanonutrition (L-Arginine Conjugated with Ag NPs) on Muscle Growth, Immune Response and Heat Shock Proteins at Different Chicken Embryonic Development Stages"

_animals, 2020, doi:10.3390/ani10040564_

Round 1

Reviewer 1 Report

L20.  Could have rather than be

L20.  Survival rate and hatching rate

Line 22.   myoD should be written out in the simple summary.

L23.  Spell out SGOT and SGPT in simple summary.

L26-27.  Awkward and not very clear

L31.  Should be “of broiler chickens”                                 

L47.  Hypertrophy and both

L66 72.  Somewhere in this section, you should present why silver nanoparticles (not just nanoparticles)

L103.  Until now?

L116.  Chemiluminescence

L117.  Falcon Labware

L123.  Watch the notation for the chemical compound

Watch temperature notation (example 90â—¦ every)

Table 1.  No need for the egg numbers if it is the same for each group

Figure 1.  Figure did not add materially to the manuscript.  Consider removing.

Figure 3.  Wavelength, not wavenumber

Figure 5.  Determinations?  Not clear.

All figures should be formatted similarly and prepared for publication.

For figure captions, suggest not using the description “small characters” but rather something similar to “Different letters among lines at each age indicate different values: a, b, c, d: p <0.05; A, B, C, D: p <0.01”

L380.  performed on muscles (Should this be on muscle tissue or extracts or similar?

L386.  Should be Fig. 10 C

References.  Should be “1. Author 1, A.B.; Author 2, C.D. Title of the article. Abbreviated Journal Name YearVolume, page range.”  Most of the references used full journal name.

Author Response

===========

Response to Reviewer 1 Comments

L20.  Could have rather than be

L20.  Survival rate and hatching rate

Ans: As per reviewer suggestion suitable sentence included in the revised manuscript.

Line 22.   myoD should be written out in the simple summary.

Ans: We have written the full form of myoblast determination protein (myoD) in the revised manuscript.

L23.  Spell out SGOT and SGPT in simple summary.

Ans: We have given the full form of serum glutamate oxaloacetate transaminase (SGOT) and serum glutamate pyruvate transaminase (SGPT) in the revised manuscript.

L26-27.  Awkward and not very clear

Ans: As per reviewer suggestion we have re-written the sentence in the revised manuscript.

L31.  Should be “of broiler chickens”    

Ans: Suitable changes made in the revised manuscript

L47.  Hypertrophy and both

Ans: We have deleted the “both” in the revised manuscript.

L66 72.  Somewhere in this section, you should present why silver nanoparticles (not just nanoparticles)

Ans: As per reviewer suggestion included the suitable literature in the revised manuscript that “Recent studies have shown that Ag NPs penetrated livers of chickens, based on their physical activity, surface ratio and chemical stability. Previous reports stated that in ovo supplementation of AgNPs could promoted growth performance and immune system of broilers and layers [Subramaniyan et al., 2018]

L103.  Until now?

Ans: Yes, until now there were no study about in ovo injection of conjugated Ag NPs with L-arginine.

L116.  Chemiluminescence

Ans: As per reviewer suggestion we have corrected the spell mistake of “Chemiluminescence” in the revised manuscript.

L117.  Falcon Labware

Ans: As per reviewer suggestion we have corrected the spell mistake of “Falcon Labware” in the revised manuscript.

L123.  Watch the notation for the chemical compound

Ans: We have included the notation of the chemical compound.

Watch temperature notation (example 90â—¦ every)

Ans: Line No:159, the “90â—¦” is replaced by “90°C”

Table 1.  No need for the egg numbers if it is the same for each group

Ans: As per reviewer suggestion deleted the “No. of Eggs” in the table 1

Figure 1.  Figure did not add materially to the manuscript.  Consider removing.

Ans: As per reviewer suggestion we have removed the Figure 1 in the revised manuscript.

Figure 3.  Wavelength, not wavenumber

Ans: It is “wavelength”

Figure 5.  Determinations?  Not clear.

Ans: determinations mean "the number of chicks (n=12)".

All figures should be formatted similarly and prepared for publication.

Ans: We have formatted the all figures are similarly in the revised manuscript.

For figure captions, suggest not using the description “small characters” but rather something similar to “Different letters among lines at each age indicate different values: a, b, c, d: p <0.05; A, B, C, D: p <0.01”

Ans: Suitable changes made in the Figure caption (Figure 9).

L380.  performed on muscles (Should this be on muscle tissue or extracts or similar?

Ans: We performed in muscle tissue and it have mentioned it in the revised manuscript.

L386.  Should be Fig. 10 C

Ans: As per reviewer suggestion suitable changes made in the revised manuscript.

References.  Should be “1. Author 1, A.B.; Author 2, C.D. Title of the article. Abbreviated Journal Name Year, Volume, page range.”  Most of the references used the full journal name.

Ans: All referenced has been checked and charged according to the journal format.

Reviewer 2 Report

Comments for author:

Overall, the study is interesting but I have few comments for further correction.

The general title can be revises as indicated below.

Effects of in ovo supplementation with nano- nutrition (L-Arginine conjugated with Ag NPs) on   muscle growth, immune response and heat shock proteins at different embryonic development stages of Chicken.

The simple summary can’t clearly represent the study story well and it looks vague, please rewrite it.

Line 254, 255 The terms wrote as “Fig. 5 and Fig.6” are not according to the journal guidelines, please see guidelines.

Author Response

Response to Reviewer 2 Comments

Overall, the study is interesting but I have few comments for further correction.

The general title can be revises as indicated below.

Effects of in ovo supplementation with nano- nutrition (L-Arginine conjugated with Ag NPs) on muscle growth, immune response and heat shock proteins at different embryonic development stages of Chicken.

The simple summary can’t clearly represent the study story well and it looks vague, please rewrite it.

ANS: As per reviewer suggestion we have improved and re-write the simple summary in the revised manuscript.

Line 254, 255 The terms wrote as “Fig. 5 and Fig.6” are not according to the journal guidelines, please see guidelines.

ANS: We have changed the terms “Figure 5 and Figure 6” according to the journal format in the revised manuscript.